# Potential Use of Cardunculus Biomass on *Pleurotus eryngii* Production: Heteroglycans Content and Nutritional Properties (Preliminary Results)

**DOI:** 10.3390/foods12010058

**Published:** 2022-12-22

**Authors:** Valerio Battaglia, Roberto Sorrentino, Giulia Verrilli, Luisa del Piano, Maria Cristina Sorrentino, Milena Petriccione, Mariarosaria Sicignano, Anna Magri, Michele Cermola, Domenico Cerrato, Ernesto Lahoz

**Affiliations:** 1Council for Agricultural Research and Economics—Research Centre for Cereal and Industrial Crops (CREA-CI), Via Torrino 3, 81100 Caserta, Italy; 2Department of Precision Medicine, Università degli Studi della Campania “L. Vanvitelli”, 80138 Naples, Italy; 3Council for Agricultural Research and Economics—Research Centre for Olive, Fruit and Citrus Crops (CREA-OFA), Via Torrino 3, 81100 Caserta, Italy; 4Department of Environmental, Biological and Pharmaceutical Sciences and Technologies—DiSTABiF, University of Campania “Luigi Vanvitelli”, Via Vivaldi 43, 81100 Caserta, Italy

**Keywords:** cardunculus, *Cynara cardunculus*, *P. eryngii*, circular economy, functional food, heteroglycan

## Abstract

The new perspective of using waste biomass to cultivate mushrooms as a source of protein for human nutrition, in line with the circular economy principles, is receiving increasing attention in the scientific community and represents great wealth in terms of environmental sustainability. *Pleurotus eryngii* is a mushroom also known as cardunculus mushroom due to its ability to grow on this plant. This study explores the potential intrinsic properties of cardunculus (for example, the presence of inulin in the roots) as raw material for the growth of cardunculus mushrooms, and the influence on heteroglycan content and nutrition parameters of the fruiting bodies. Both mycelium and fruiting bodies were used to determine the heteroglycan content in the presence of inulin or cardunculus roots rich in inulin. To produce heteroglycans from *P. eryngii* in greater quantities and shorter times without having to wait for the formation of the fruiting bodies, the mycelium could be used. The results showed that the presence of cardunculus biomass positively influences the heteroglycan content of *P. eryngii*. In terms of nutritional parameters, higher contents of polyphenols, flavonoids, anthocyanins, and antioxidant activity were detected in *P. eryngii* grown on the cardunculus stem and root substrate. In conclusion, recycling cardunculus biomass to generate growth blocks for edible mushrooms is a winning choice due to the opportunity to use this biomass waste, which is gaining more and more attention due to the increase in cultivated areas and the use of fruiting bodies of *P. eryngii* as a functional food and source of molecules with potential biological activities.

## 1. Introduction

Mushrooms are organisms consumed as food all over the world and are appreciated for their organoleptic, health qualities for human dietary needs, and source of proteins. Among them, many species are commercially available as supplements; moreover, the metabolites obtained from the mycelia and fruiting bodies of some fungal species have multiple pharmacological activities [1].

Among the crops to be used for this purpose, *Cynara cardunculus* L. var. *altilis* (cardunculus) may play a prominent role since it is well known for multiple uses including energy and biofuels, cellulosic pulps, phytochemicals, and food and feed [2].

The cultivation in Italy of cardunculus represents a rich opportunity since it is possible to obtain various bioproducts (bioplastic, biopesticides, biocosmetics, bioenergy, etc.) [3] and use the biomass (stems and roots with high inulin content) for edible mushroom cultivation. Growing mushrooms from cardunculus waste, such as stem and root biomass after the harvesting of flower heads, to use seeds for the production of several byproducts drives down costs of transporting biomass (mostly originating from abroad) and makes the implementation of a circular economy feasible [4]. Two of the most important byproducts currently used in Italian agriculture are bioplastic for organic mulch and pelargonic acid as weed and/or sucker control. Cardunculus roots are well known in the literature as a source of inulin (>40%) [5,6], and in southern Italy, the wild *Pleurotus eryngii* (DC.) Quél. is commonly associated with cardunculus (cardo in the Italian language), hence its Italian common name is “cardoncello” [7]. In Italy, molasses that contains sucrose is the carbon source to grow *P. eryngii*. Molasses is usually imported from other countries. *Pleurotus eryngii* has a high affinity to inulin, which may allow us to replace molasses with cardunculus roots.

Cultivable mushrooms are grown successfully on different substrates such as cereal meals, brans, straw, sugar beet, etc., but their yield and chemical characteristics are strongly influenced by the composition of waste biomass [8].

The possibility to grow on various agro-residues offers an interesting prospect for converting different types of wastes [9] and/or lignocellulose residues into high-quality proteins, characterized by a complete amino acid and vitamin composition [10]. In fact, the basidiomycetes fungi belonging to the genus Pleurotus have an elevated protein content with a valuable essential amino acid scoring pattern and are characterized by a unique dietary fiber profile (mainly comprised of branched β-glucan), high levels of some B group vitamins, vitamin D, Fe, Zn, Cu, Se, and some bioactive compounds. Therefore, their sodium and fat contents are low, and they are also known to possess very important and advantageous medicinal properties such as anti-cancer, anti-microbial, anti-diabetic, anti-hypercholesterolemic, and immunomodulating activities, pulmonary cytoprotective effects, and even implications for coronavirus disease (COVID-19) immunotherapies [11,12,13,14,15,16,17,18,19,20]. Due to these features, the genus Pleurotus could make an important contribution to sustainable functional food design. Furthermore, *P. eryngii* is reported as the species, in the genus Pleurotus, with the highest glucan content, which could be used as a nutrient source of high glucan production for the modern food industry [21].

Glucans (C_6_H_12_O_5_) are polysaccharides consisting of glucose monosaccharides linked by α, β, or mixed α-β glycosidic bonds; they are structural components of the cell walls of the fungal mycelium and are known for their immune-modulating activity [17].

In a previous study, heteroglycans (HGlyc) were extracted in large amounts from the mycelium of *Ganoderma lucidum* (Curtis) P. Karst. avoiding the need to obtain fruit bodies, which take longer to grow [15]. In the present work, both the production of the mycelium and fruiting bodies were evaluated to verify in which conditions the heteroglycan content was higher.

*Pleurotus eryngii* is one of the most important cultivated mushroom species with a high ability to colonize lignin and a high affinity for cardunculus plants.

The cardunculus biomass is rich in biotin and thiamin, two vitamins recommended by Chang et al. [22] to be incorporated into the substrate to enhance fruit bodies. In addition, cardunculus contains a high quantity of inulin since it belongs to a species of the Asteraceae family with an interesting chemical composition, as well as in terms of the C/N ratio [23]. For these reasons, cardunculus biomass is strictly advised to be used for the cultivation of *P. eryngii* mycelia and fruit bodies.

The objectives of the present study were:(i)To evaluate the use of cardunculus biomass on *P. eryngii* fruiting bodies yield.(ii)To evaluate the possibility to use *P. eryngii* mycelium, cultivated on media rich in inulin from cardunculus roots, as a source to produce functional heteroglycans.(iii)To evaluate the influence of cardunculus biomass on heteroglycan content in fruit bodies of *P. eryngii*.(iv)To assess the nutraceutical parameters of fruiting bodies grown on cardunculus biomass.

## 2. Materials and Methods

### 2.1. Fungal Isolates

Five isolates of *P. eryngii* (Pe1Ce ÷ Pe5Ce) were collected from five industrial cultivations of *P. eryngii* in the Potenza province, Basilicata region, in Southern Italy, and molecular characterization by Internal Transcribed Spacer (ITS) sequencing was carried out to ensure that isolates belonged to the *P. eryngii* species. The universal primers used for fungal amplification were ITS1 (5′TCC GTA GGT GAA CCT GCG G 3′) and ITS4 (5′TCC TCC GCT TAT TGA TAT GC 3) [24]. The amplification reaction mixture consisted of DNA (15 ng), ddH_2_O, 10X Dream Taq Buffer, 1 mM MgCl_2_, 0.2 mM each dNTP, 0.6 mM primer, and 1 U Dream Taq DNA polymerase (Thermo Fisher Scientific, Monza, Italy). The PCR was carried out as follows: Initial denaturation at 95 °C for 5 min, followed by 35 cycles of denaturation at 95 °C for 30 s, annealing at 55 °C for 1 min, extension at 72 °C for 1 min, and final extension at 72 °C for 6 min. Each amplified product was separated in a 2% agarose gel in 1× Tris-borate-EDTA buffer and visualized using SYBR^®^ Safe with blue-light. PCR products were purified and sequenced by BMR Genomics Srl (Padova, Italy). The resulting sequences were trimmed and then subjected to BLASTn analysis.

### 2.2. Spawn-Production

Millet seeds were used for spawn stock used for all the experiments. Three kilograms of millet for each isolate were weighed and washed with tap water and soaked in water plus 10 g L^−1^ of Czapek dox broth (Oxoid) for 24 h at 4 °C. Furthermore, 500 g of millet seeds was packed in autoclavable bags and sterilized in an autoclave at 121 °C for 90 min. Bags containing the sterile substrate were cooled and inoculated aseptically by transferring 10 plugs, 1 cm in diameter, of 10-day-old fungal cultures and then incubating them in a chamber at 22 °C until the millet was completely colonized by mycelia (approximately 20 days). These preparations were stored at 4° C and used for all the experiments performed.

### 2.3. Assessment of the Ability of P. eryngii to Grow on Cardunculus Roots and Stems

To assess the ability of the five *P. eryngii* isolates to grow on cardunculus roots or stems, three media (wheat straw from cv. Marco Aurelio: WS, Cardunculus roots: TR, Cardunculus stems: TS) were prepared with 2% water agar and amended with 3 g L^−1^ CaCO_3_. The WS medium contains 20 g L^−1^ of wheat straw with a carbon–nitrogen (C/N) ratio of 59, plus 0.1% NH_4_Cl so nitrogen was not a limiting factor for Pleurotus growth. Medium TR and TS had a C/N of 31 and 25, respectively. The materials (wheat straw, cardunculus roots, or stems) rehydrated to 60–65% of humidity were homogenized to obtain a uniform size and distribution of particles inside the media. Neither vitamins nor carbohydrates were added. The media had a pH of 6.5–6.8. For each of the five isolates, four Petri dishes were inoculated in the middle with two millet seeds taken from spawn stocks described in 2.2. The radial growth was measured 5–9–12 days after the inoculation of Petri dishes, and the last assessment was made when at least one isolate reached the edge of Petri dishes in one of the three media. The experiment was performed twice. At the end of the experiments, the isolates with the best performance on cardunculus were chosen for further experiments.

### 2.4. Mycelial Growth

To assess the best carbon source to increase the biomass production of *P. eryngii* mycelium, seven different concentrations of inulin and/or sucrose as the carbon source were compared (Table 1). The media were prepared using Czapek salts; two-liter flasks for each of the seven combinations of inulin/sucrose were inoculated in triplicate with 5 mm diameter plugs of 7-day-old fungal colonies of the Pe1Ce isolate because, as stated in Section 2.3, these had the best growth results on cardunculus substrates. Flasks were then incubated in a chamber at 22 ± 1 °C in the dark. After 14 days, the mycelium was collected and weighed. Then the samples were dried to calculate the Dry Weight (DW). The results were the means of the three experiments with four replicates.

### 2.5. Evaluation of Fruiting Bodies Production Caractheristics on Different Subtrates

To assess the fruiting bodies yield and the heteroglycans content, the following four different substrates were prepared: TSI: 600 g of cardunculus stems rehydrated at 65–70% using water amended with 30 g of inulin and 20 g of CaCO_3_; TSS: 600 g of cardunculus stems rehydrated with water amended with 30 g of sucrose and 20 g of CaCO_3_; TSR: 400 g of cardunculus stems plus 200 g of cardunculus roots rehydrated with water amended only with 20 g of CaCO_3_; and as a control, (industrial standard) WSM: 600 g of wheat straw plus 30 g of molasses rehydrated with water containing 20 g of CaCO_3_ and 0.05% NH_4_Cl. The substrates were put in an autoclavable Sun bag with a pore size of 0.02 μm (Merck^®^) with a working volume of 600 grams.

After 24 h of soaking, the excess water was removed from 600 g of the soaked substrate by decanting. The substrates were weighed again, and the moisture content was determined. The final moisture content was measured by drying the substrate at 105 °C, and the process ended when the weight was constant after two consecutive days.

The water content was calculated by the equation:


Humidity % = (Fresh Weight − Dry Weight)/Fresh Weight × 100
(1)


All four substrates had a moisture content ranging from 72% to 78%. The substrates were autoclaved for 60 min at 121 °C in plastic bags. Then 50 g of colonized millet with the selected isolate *P. eryngii* Pe1Ce was used to inoculate each bag. Inoculated bags (four for each substrate) were then incubated at 23 °C in the dark for the days needed to complete the colonization for each substrate. After colonization, the bags were opened on top of the Sun bag. The culture room was provided with light from fluorescent bulbs with an intensity of 400 lux for 8 h of light and 16 h of dark [25]. The culture was constantly wet to maintain the required relative humidity (75–90%). Cultures were irrigated by spraying water once or twice a day. After the time needed to completely colonize bags and differentiate fruiting bodies, the number and yield of fruiting bodies were assessed. The following production parameters were measured: Mycelial Colonization (MC), Primordia Appearance (PA), and Fruiting Bodies Formation (FBF) expressed in days. To determine the weight of fruiting bodies expressed in grams, *P. eryngii* fruit bodies were harvested when the mushroom caps began to flatten out by cutting the stem at the substrate level. The Biological Efficiency (BE) index was calculated as reported by Familoni et al. [26]. The assay was repeated three times.

### 2.6. Heteroglycans Determination in Mycelium and Fruiting Bodies

Heteroglycan extraction was carried out following the protocol illustrated by Carrieri et al. [15]. The heteroglycan concentration was determined using the Congo red colorimetric method according to Nitschke et al. [27]. The calibration curve and its equation (Figure 1) were determined using the molecules obtained in our previous study [13]. Absorbance was calculated as differences between Congo red plus heteroglycan at different concentrations and Congo red alone. The concentrations reported are the means of two independent experiments with three technical replicates.

The content of heteroglycans obtained from mycelium and fruiting bodies of *P. eryngii* was compared with that obtained from mycelium of *G. lucidum* reared as reported in Carrieri et al. [15].

### 2.7. Bioactive Compounds, Antioxidant Activity, and Enzymatic Assays

Extraction was performed following the procedure described by Kaur et al. [28] on fruiting bodies of *P. eryngii* grown on TSR compared with WSM. The total phenolic content was determined according to Folin–Ciocalteu’s method [29]. The results were expressed as mg of gallic acid equivalent (GAE) per 100 g of dry weight (GAE 1g DW). The total flavonoid content was determined with the aluminum chloride colorimetric method according to Goffi et al. [30]; the resulting data were expressed as mg of catechin equivalent (CE). The total monomeric anthocyanins content was estimated with the pH-differential method reported by Adiletta et al. [31] and the results were expressed as the cyanidin-3-glucoside equivalent (C3G). The antioxidant activity was measured using the 1,1-diphenyl-2-picryl-hydrazil (DPPH) method described by Magri et al. [32], with some modifications. The assay mixture consisted of 75 µL of the extract and 1425 µL of the solution of DPPH dissolved in methanol. After 10 minutes of dark incubation, the decrease in absorbance at 515 nm was recorded and expressed as µmol Trolox equivalent (TE) g^−1^ of DW.

Polyphenol oxidase (PPO) (EC.1.10.3.1) activity was determined according to Caracciolo et al. [33]. The crude enzyme extract was obtained using 0.1 g of mushroom tissue homogenized in 5 mL of 0.2 M sodium phosphate buffer pH 6.5 with 0.125 g PVPP. The extract (10 µL) was incubated with a solution of 0.5 M catechol in 0.1 M sodium phosphate buffer at pH 6.4 in a final volume of 1.5 mL, and PPO activity was registered at 398 nm and expressed as nmol g^−1^ DW.

Lipoxygenase (LOX) (EC 1.13.11.12) activity was detected following the method described by Adiletta et al. [34], with slight modifications. The extract was obtained by homogenizing mushroom tissue powder (100 mg) with 3 mL of 50 mM potassium phosphate buffer pH 7.8 with 1 mM sodium-EDTA pH 7 and 2% (*w*/*w*) PVPP. The assay mixture consisted of 0.1 M sodium phosphate buffer pH 6, linoleic acid sodium salt 5 mM, and 50 µL of a crude enzyme extract in a final volume of 1.5 mL. LOX activity was registered at 234 nm, observing the formation of hydroperoxides, and the results were expressed in nmol g^−1^ DW. Total soluble protein content was determined by the Bradford assay [35].

### 2.8. Statistical Analysis

Each value represents the mean ± Standard Error (SE) from three independent experiments carried out in triplicate. The test for the homogeneity of variance was applied to evaluate whether the data could be combined as one experiment. Statistics were performed using GraphPad InStat version 3.00 for Windows (GraphPad Software, San Diego, CA, USA, www.graphpad.com), and differences between treatments were considered significant at *p* ≤ 0.05. The data obtained were subjected to an analysis of variance (ANOVA) for quantitative variables, and means were separated using Tukey’s test.

## 3. Results

### 3.1. Assessment of P. eryngii Isolate’s Ability to Colonize Cardunculus Roots and Stems

The present work was carried out using a single isolate of *P. eryngii*, selected on the basis of the colonization capacity of the substrates containing cardunculus biomass (WS, TR, TS) (cfr. par. 2.3). The initial results pointed out that three isolates (Pe2Ce, Pe4Ce, and Pe5Ce) (Table 2) showed a lower growth rate on all three media, indicating they are not suitable to obtain fast growth. The isolates Pe1Ce and Pe3Ce grew faster on the media WR, TS, and TR. Moreover, Pe1Ce had significantly faster growth than Pe3Ce on both TR and TS media. Based on these results, Pe1Ce was chosen to perform all of the following experiments.

### 3.2. Fungal Isolate

Through the alignment of the sequences, it was confirmed that all five isolates belonged to the *P. eryngii* complex according to De Gioia et al. [36]. After the assessment of the isolates’ affinity to the cardunculus biomass, the sequence of the chosen fungal isolate (Pe1Ce) demonstrated that our isolates belonged to *P. eyngii* specie. The sequence was deposited in GenBank with the accession number OM541308.

### 3.3. Effects on Mycelial Production by the Addition of Inulin and Sucrose In Vitro

In order to identify the best ratio between inulin and sucrose in the substrates for the growth of *P. eryngii* (Pe1Ce), seven combinations were tested. The experiment showed that the addition of inulin and sucrose in vitro affected the development of the mycelium; high inulin concentrations significantly improved the DWs of *P. eryngii* mycelium (Table 3).

### 3.4. Fruiting Bodies Yield and Heteroglycans Content on Different Substrates

Different substrates affected the mycelial growth, TSR and TSS completed the mycelial colonization in 15.6 days, while TSI and WSM took 17.4 and 19.2 days, respectively (Table 4). Regarding the development of the primordia, *P. eryngii* requires 22 days to obtain the complete first primordium of fruiting bodies in the TSI substrate, while on the substrates without inulin, it takes a minimum of 26 days. The formation of the fruiting bodies and the yield are influenced by the growth substrate as reported in Table 4. The formation of the fruiting bodies is faster on TSR (5 days vs. 7 and 8 days) compared to other substrates. The yield is positively influenced by the presence in the substrate of cardunculus root since the mean weight of fresh mushrooms was the highest in TSR (371 g) compared to the other substrates. The biological efficacy index (BE) confirms TSR as the substrate with the highest BE rate (61.9 %) among all the tested substrates.

### 3.5. Heteroglycans Content in Fruiting Bodies and Mycelium of P. eryngii

As reported in Figure 2, the presence of inulin in mycelium growth and cardunculus biomass in the obtained fruiting bodies affects the content of heteroglycans in *P. eryngii*. In this study, we observed that in the mycelium of *P. eryngii,* the quantity of heteroglycans is higher (1706.3 µg g^−1^) when inulin is added to the substrate (739.7 µg g^−1^) compared to the sucrose alone. In particular, the content of heteroglycans is also higher if compared to *G. lucidum* (468.2 µg g^−1^) [15].

The fruiting bodies of *P. eryngii* cultivated on the TSR substrate showed higher heteroglycan content (1037.8 µg g^−1^) compared to the control substrate WSM (833.4 µg g^−1^), likely due to the absence of cardunculus.

### 3.6. Evaluation of Different Bioactive Compounds and Antioxidant Activity and Enzymes Involved in Browning Reaction and Membrane Damage

Bioactive compound content and the antioxidant activity of fungi samples are shown in Figure 3.

In this study, WSM samples showed the lowest value of polyphenols and flavonoids compared to TSR (Figure 3A,B). The anthocyanins content in the TSR sample amounted to 22.6 ± 1.30 mg C3G 100 g^−1^ DW compared to 9.59 ± 0.89 mg C3G 100 g^−1^ DW of WSM one (Figure 3C). Significant differences were also highlighted in antioxidant activity with the highest value in TSR (21.35 ± 1.43 µmol TE g^−1^ DW) compared with the WSM sample (18.11 ± 0.36 µmol TE g^−1^ DW) (Figure 3D). The results of enzyme activities related to the browning reaction and oxidative damage in the two fungi samples are shown in Figure 4 In the present study, the polyphenoloxidase activity of WSM samples resulted in 2-fold lower values than TSR samples (Figure 4A). The same trend was recorded for lipoxygenase. In particular, the lipoxygenase activity in the WSM sample was 0.07 ± 0.01 nmol g^−1^ DW, compared to 0.24 ± 0.06 nmol g^−1^ DW in the TSR samples (Figure 4B).

## 4. Discussion

*Pleurotus eryngii* belongs to the genus Pleurotus in the family of Pleurotaceae [37]. It has been attracting more and more attention recently due to its nutritional properties and the possibility of growth on alternative substrates. The cardunculus, known as *C. cardunculus,* is a very versatile plant with various applications [38]. In the Mediterranean region, cardunculus is used for human nutrition, as a quality vegetable, and for the coagulation of milk as a substitute for rennet. Furthermore, the use of its aerial biomass for energy purposes has been receiving increasing interest due to its characteristics such as adaptation to Mediterranean climates and high potential yields. The biomass of the cardunculus can also be exploited as a solid biofuel or to produce paper pulp and oilseeds, useful in the production of edible oil or biodiesel [38]. Innovative use of cardunculus relates to its waste biomass as a substrate for the cultivation of edible mushrooms. This latter is a key factor for the conversion of low-value waste into a higher-value commodity as an alternative food source for humans (proteins and functional molecules). Exploiting this feature and from the perspective of a circular economy of reuse of residual biomass from agriculture, in the present study, we evaluated the affinity, growth rate, and nutritional properties of *P. eryngii,* cultivated by making use of cardunculus biomass as a growth substrate. Since De Gioia et al. [36] described that the gene pool of Pleurotus is very variable in morphological and molecular traits and that variation within groups is larger than between groups, with the genetic diversity detected within groups likely due to an efficient gene flow and high genetic compatibility, we aimed to evaluate the growth rate of our *P. eryngii* isolates. The first step of our work was, indeed, the selection of *P. eryngii* isolates on the basis of their ability to grow on substrates containing roots and stem of cardunculus in comparison to the growth on the common substrate containing straw and molasses (cfr. par. 3.1). The results of in vitro biomass production tests showed the isolate Pe1Ce as the best *P. eryngii* isolate to carry out further experiments.

Recently, the idea to develop functional foods or drugs containing functional fungal polysaccharides has been gaining great attention [39]. It is known that the use of inulin and sucrose as carbon sources allows the faster growth of fungi [5,6,40]. Our results demonstrated that inulin is a suitable carbon source to replace the use of molasses. Our results confirmed previous data reported in the literature; in particular, the substrates with high inulin concentration (media 6 and 7) allowed us to obtain a larger amount of fungal mycelium. This study has highlighted that higher heteroglycan production in the mycelium and fruiting bodies can be obtained using a medium with inulin.

In our work, we used the mycelium of *G. lucidum* as a term of comparison since this microorganism is widely used on an industrial scale for the production of beta glucan supplements [40]. This fungus has potent pharmacological properties due to its heteroglycan content [13,35,36]. In Carrieri et al. [15], the possibility of growing *G. lucidum* in a bioreactor to produce mycelium and extract water-soluble heteroglycans and heteroglycans was successfully verified. In the present work, the heteroglycan content obtained from *G. lucidum* mycelium is considerably lower when compared to *P. eryngii*. Our results also indicate that *P. eryngii* mycelium is an excellent producer of heteroglycans, producing them rapidly and in large amounts.

In recent years, the residual biomass of cardunculus increased its availability in the environments of southern Italy; therefore, we used this biomass in our work to cultivate *P. eryngii* to obtain fruiting bodies as a functional food. Cardunculus crop residues are also a natural source of inulin in roots [41], hence we chose, in line with the principles of the circular economy, to use stems and roots as a lignin-cellulose source and an inulin source, respectively, for the growth of the basidiomycota *P. eryngii.* Avni et al. [21] reported *P. eryngii* as the Pleurotus species with the highest glucan content that could be used as a nutrient source of high glucan production.

This is the first report of *P. eryngii* cultivated on cardunculus residues; in the literature, *Pleurotus* spp. were cultivated by using several types of waste such as coffee, sawdust, and sugarcane bagasse [42], on a rice straw substrate, and in banana straw, as reported in [43]. In Tarko et al. [42], *P. eryngii,* grown on coffee waste, showed a mean weight of fruiting bodies of 318 g, while in this work, the best growth substrate (TSR) allowed us to reach 371.7 g.

The fruiting body of *P. eryngii* is a commercial product as it is and, therefore, represents the main goal of this work. Fruiting bodies showed high organoleptic and nutritional properties when *P. eryngii* was grown on cardunculus residues (TSR). In particular, fruiting bodies displayed high bioactive compounds and antioxidant activity and enzymatic activities and a greater amount of heteroglycan compared to WSM. Fungi have many nutritional and nutraceutical properties, in addition to antimicrobial, antioxidant, antitumor, and anti-inflammatory activities [44]. Mushrooms’ chemical composition is variable in relation to the isolates, species, growing methods, environmental conditions, mother cultures, harvest time, and storage conditions [11]. The substrate nature and chemical composition significantly influence the accumulation of bioactive compounds and antioxidant properties in the fruiting bodies of *P. eryngii* [45,46]. Cotton waste resulted in the highest amount (230.45 mg GAEs kg^−1^ of DW) of TPC, while the lowest amount (137.08 mg GAEs kg^−1^ of DW) was detected from mushroom fruiting bodies grown on sawdust [45]. However, these values were lower compared to our results. High TPC values might be due to the maximum supply of nutrients from the substrate, which improves the non-enzymatic antioxidant system involved in the removal of free radicals and the activation of antioxidant enzymes [47]. Isam et al. [48] reported a TPC for *P. citrinopileatus* and *P. eryngii* of 2.72 and 3.61 mgGAE g^−1^, respectively. Gąsecka [49] reported a TPC of 7.91 ± 1.02 mg of chlorogenic acid per g DW and flavonoid content equal to 1.26 ± 0.17 mg of rutin equivalents per g of DW. Furthermore, in another study, the flavonoid content reported was 0.9 ± 0.1 mg of quercetin 100 g^−1^ FW, and DPPH activity reached values of 40.6 ± 0.3% [50]. The content of phenolic compounds is an important indicator of antioxidant capacity, with a close relationship between two these variables. *P. eryngii* displayed higher antioxidant activity in terms of radical scavenging activity on the DPPH free radical compared to other oyster mushrooms [51]. Mushroom pigments, produced by the polymerization of phenolic compounds, are synthesized by two biosynthetic pathways, namely, the acetate–malonate pathway and the shikimic acid pathway. The anthocyanin responsible for the brown-colored skin of fungi was principally melanin [52]. PPO is a key enzyme that participates in tissue browning [53]. In this study, a lower PPO activity highlights a reduction of tissue browning. In recent years, several strategies have been tested to inhibit PPO activity in postharvest *P. eryngii* [53,54]. Furthermore, a low LOX activity suggests reduced lipid oxidation on cell membranes after harvesting [55].

The nutritional parameters evaluated indicate that the choice of growth substrate is suitable for obtaining fruiting bodies with a high nutraceutical and functional value. The novelty of this study is represented by the use of a crop residue with a biochemical composition of the biomass that covers all the nutritional needs of *P. eryngii,* in addition to the presence of a high content of inulin in cardunculus roots.

Using TSR, we obtained an increment of heteroglycan content of 20.3% compared with the traditional growing substrate WSM.

All reported results indicate that cardunculus biomass residue is an excellent choice as a growing medium for edible *P. eryngii* mushrooms. The fruiting body of *P. eryngii* represents a potential functional food that can be used as a component of a rich and varied diet. Further investigations on the biochemical role of inulin in the growth of *P. eryngii*, the selection of new, more productive strains of cardunculus, and the large-scale development of a bioreactor for mycelium production will be needed.

## Figures and Tables

**Figure 1 foods-12-00058-f001:**
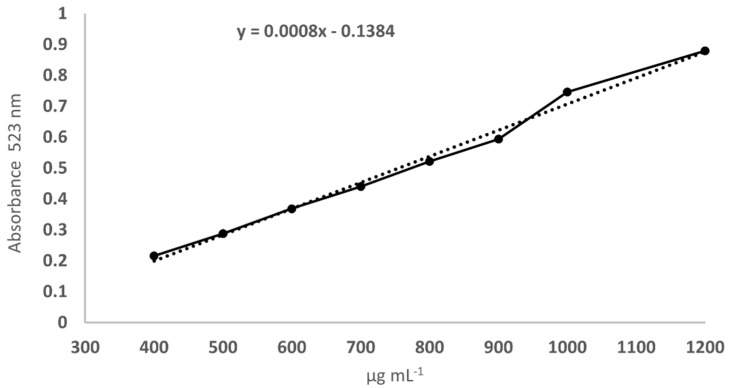
Calibration curve for Congo red method and its equation.

**Figure 2 foods-12-00058-f002:**
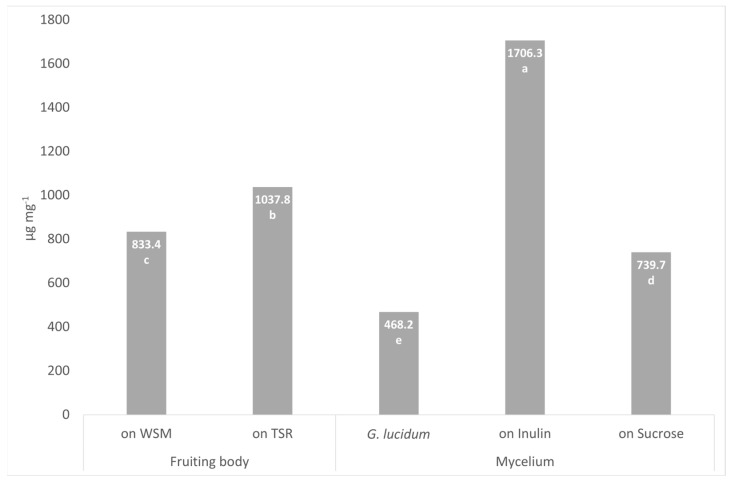
Heteroglycan content in fruiting bodies grown on wheat straw molasses (WSM) and cardunculus stems and cardunculus roots (TSR), and mycelium of *P. eryngii* and *G. lucidum.* The data represent the mean values of three experiments with three technical replicates. Different letters above the bars indicate significantly different according to Tukey’s test at *p* ≤ 0.05.

**Figure 3 foods-12-00058-f003:**
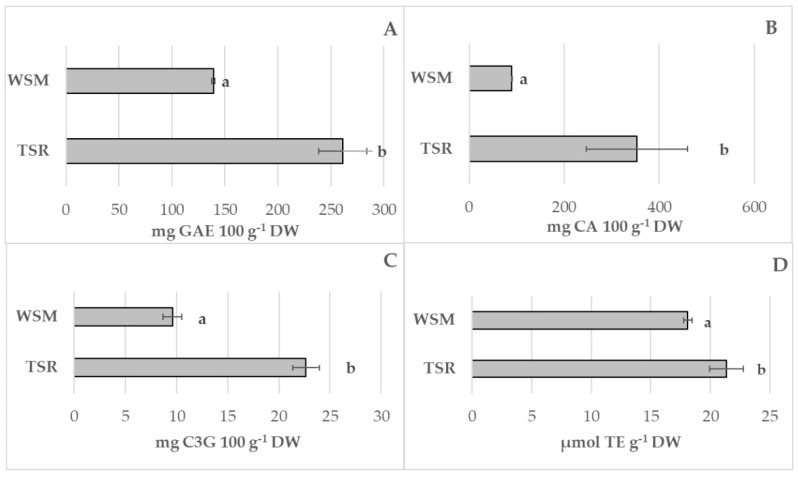
Total polyphenol (**A**), total flavonoids (**B**), total anthocyanin (**C**), and antioxidant activity (**D**) in *P. eryngii* obtained by two different substrates (WSM and TSR). The data represent the mean values of three experiments with three technical replicates. Means with the same letter are not significantly different according to Tukey’s test at *p* ≤ 0.05.

**Figure 4 foods-12-00058-f004:**
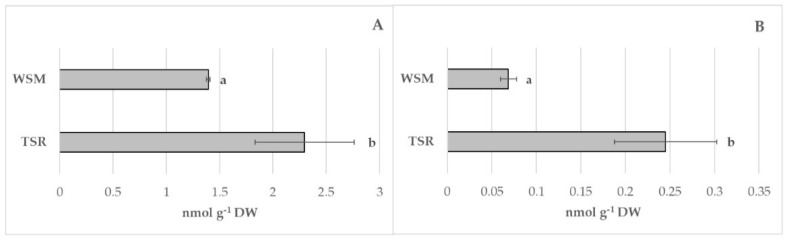
Polyphenoloxidase (**A**) and lypoxigenase activity (**B**) in *P. eringii* obtained by two different substrates (WSM and TSR). The data represent the mean values of three experiments with three technical replicates. Different letters aside the bars are significantly different according to Tukey’s test at *p* ≤ 0.05.

**Table 1 foods-12-00058-t001:** Grams of inulin and sucrose in the media.

Media	Sucrose (g L^−1^)	Inulin (g L^−1^)
1	30	0
2	27	3
3	21	9
4	15	15
5	9	21
6	3	27
7	0	30

**Table 2 foods-12-00058-t002:** Speed growth expressed in mm day^−1^ of the isolates Pe1Ce to Pe5Ce grown on the media.

Substrates ^1^	Isolates					Mean
	Pe1Ce	Pe2CE	Pe3Ce	Pe4CE	Pe5Ce	
WS	0.6	0.4	0.5	0.4	0.4	0.48
TR	0.7	0.5	0.6	0.5	0.5	0.58
TS	0.8	0.5	0.7	0.5	0.6	0.62
Mean	0.68	0.48	0.62	0.49	0.52	
Statistical analysis ^2^						
	ANOVA			LSD 0.09		
Isolate	**			0.062		
Substrate	**			0.075		
Isolate × Substrate	**			0.087		

^1^ (WS) Wheat straw having C/N of 59 and 0.1% NH_4_Cl; (TR) Cardunculus roots having a C/N of 31; (TS) Cardunculus stems having a C/N of 25. ^2^ The data represent the mean values of three experiments with four technical replicates. The variances between the three experiments were not significant. Parameters with ** are statistically different at *p* ≤ 0.05.

**Table 3 foods-12-00058-t003:** Mycelial yield in liquid media by adding different concentrations of sucrose and inulin.

Media	Carbon Source		DW ^1^ after 12 Days of Growth (g)	
	Sucrose	Inulin		
	(g L^−1^)	(g L^−1^)		
1	30	0	10.1 ± 1.72	b
2	27	3	8.9 ± 1.12	b
3	21	9	9.1 ± 1.24	b
4	15	15	10.3 ± 1.45	b
5	9	21	9.2 ± 1.47	b
6	3	27	16.1 ± 1.87	a
7	0	30	15.9 ± 2.11	a

^1^ DW = Dry Weight and the data represent the mean values of three experiments with three technical replicates. Data indicate mean ± SE. Means with the same letter are not significantly different according to Tukey’s test at *p* ≤ 0.05.

**Table 4 foods-12-00058-t004:** Influence of different substrates containing cardunculus residues on fruiting bodies yield and production parameters.

Substrate ^1^	MC ^2^ (days)	PA ^3^ (days)	FBF ^4^ (days)	MW ^5^ (g)	BE ^6^ (%)
TSI	17.4 ± 0.82 b	22.8 ± 1.12 c	7.0 ± 0.6 a	305.7 ± 12.5 b	50.9 ± 5.7 bc
TSS	15.6 ± 0.67 c	26.5 ± 0.98 b	7.0 ± 0.5 a	310.5 ± 9.4 b	51.7 ± 7.9 b
TSR	15.6 ± 0.82 c	26.0 ± 0.89 b	5.0 ± 0.4 b	371.7 ± 11.2 a	61.9 ± 4.7 a
WSM	19.2 ± 1.02 a	29.3 ± 1.21 a	8.0 ± 0.6 a	280.6 ± 9.8 c	46.7 ± 4.8 c

^1^ Substrate: TSI: 600 g of cardunculus stems rehydrated at 65–70%, added with 30 g of inulin and CaCO_3_; TSS: 600 g of cardunculus stems rehydrated and amended with 30 g of sucrose and CaCO_3_; TSR: 400 g of cardunculus stems plus 200 g of cardunculus roots amended with CaCO_3_; WSM: 600 g of wheat straw plus 30 g of molasses, CaCO_3_, and NH_4_Cl. ^2^ Mycelial Colonization: Days necessary to the fungus to colonize the entire Petri dish (Ø = 90 mm) ^3^ Primordia Appearance: Days from the inoculation of the agar plug until the development of the fruiting bodies primordia. ^4^ Fruiting Bodies Formation: Required days to observe the complete development of fruiting bodies after the primordial appearance. ^5^ Mean Weight of fruiting bodies. ^6^ Biological Efficiency of substrate. The data represent the mean values of three experiments with three technical replicates. Means with the same letter are not significantly different according to Tukey’s test at *p* ≤ 0.05.

## Data Availability

The data used to support the findings of this study can be made available by the corresponding author upon request.

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
