# Peer review of "Potential Use of Cardunculus Biomass on Pleurotus eryngii Production: Heteroglycans Content and Nutritional Properties (Preliminary Results)"

_foods, 2022, doi:10.3390/foods12010058_

Round 1
Reviewer 1 Report
I think that the manuscript entitled “Potential use of thistle biomass on Pleurotus eryngii production: heteroglycans content and nutritional properties" deserves publication in the Foods after major revision.
The manuscript is interesting. However it focuses mainly on modifying Pleurotus eryngii cultivation (which is, of course, a food product). The part concerning food analysis is incomplete and has the nature of an additive or subject of research, the more the only research methods concerning this part are quite old. In addition the size of SE for sample with thistle raises great concern as to the reliability of the statement made. If the Editor agrees, the manuscript can be sent after completing the food research on all grown Pleurotus eryngii and the corrections already presented in the work.
The review is also very difficult due to the lack of numbering of the pages and the lines.

Author Response
Reviewer 1:
I think that the manuscript entitled “Potential use of thistle biomass on Pleurotus eryngii production: heteroglycans content and nutritional properties" deserves publication in the Foods after major revision.
The manuscript is interesting. However it focuses mainly on modifying Pleurotus eryngii cultivation (which is, of course, a food product). The part concerning food analysis is incomplete and has the nature of an additive or subject of research, the more the only research methods
concerning this part are quite old. In addition the size of SE for sample with thistle raises great concern as to the reliability of the statement made. If the Editor agrees, the manuscript can be sent after completing the food research on all grown Pleurotus eryngii and the corrections
already presented in the work.
The review is also very difficult due to the lack of numbering of the pages and the lines.
The present work focused primarily on the increasing of heteroglycan and nutraceutical content of P. eryngii that is the most cultivated mushroom species in the same areas where thistle is cultivated,
hence its biomass is highly available.
The data reported agreed with the improvement of nutraceutical features obtained by cultivating the mushroom on thistle. Thus, Pleurotus grown on thistle as mother culture increased the content in
heteroglycans, polyphenol, flavonoids, etc... compared with Pleurotus grown on traditional WSM (wheat straw molasses). We propose, for the human consumption, P. eryngii fruiting bodies obtained by the proposed method, to replace P. eryngii cultivated by traditional method.
Concerning the standard error, we would like to specify that these values must be related to the type of experiment carried out. In other papers, in which the same cultivation method was applied (Kirbag
and Akyuz 2008; Panjikkaran and Mathew 2013), the standard error in percentage is in line or higher than those obtained in our experiment, since the variables involved are comparable to those of semifield experiment.
Reviewer 2 Report
The study investigates the possibility of using thistle biomass to grow thistle mushrooms (Pleurotus eryngii), with the hypothesis essentially being that thistle makes thistle mushrooms grow better, possibly due to thistle’s chemical content (e.g. inulin). This involves studying the growth and biochemical properties of the resultant mushrooms. The project conceptually makes sense, and the general straight-forwardness of the project’s intent and experimental design make for a well-contained and theoretically easily-understood
There are of course, certain things to address:
1) Inulin is a common compound, and certainly not unique to thistle; nor was it specified in the publication that thistle root has, for example, exceptionally high levels of inulin, that would justify selecting only inulin as a supplement. Given inulin’s widely-reported function as a useful substrate for microbes/as a prebiotic, it isn’t surprising that it would have some impact on fungal growth as well.
In other words; what other compounds are present in thistle (either unique to thistle, or in exceptionally high levels) that could potentially be studied? Biotin and thiamine were mentioned in the introduction, but were not tested.
2) Not clear why the authors say that “Pleurotus eryngii is one of the most important cultivated mushroom species”. What is especially important about this species?
3) For section 2.5, how long were the samples subjected to 105â—¦C? And how did the authors ensure that the sample was completely dried? E.g. were samples reweighed at least twice for consistency?
4) For the statistical analysis in Figure 2, presumably the authors are comparing between the five strains within each growth media (i.e. all three growth media are independently evaluated). Might also be worth analyzing for significant differences in growth for the same strain but different growth media.
5) I personally have reservations about the use of only a triplicate when evaluating samples like this. Even though many have done it, in my personal experience having a larger number of replicates (e.g. five) gives more reliable information. This is especially so for when error bars are quite large (e.g. some of the columns in Figures 4 and 5).
6) Even within the same column, the number of significant figures/decimal points are sometimes inconsistent. For example, “7 ± 0.62”.
7) The formatting of CaCO3 in the Table 3 legend is inconsistent.
8) Figure 3 is missing indicators of replication, error bars, and statistical analyses. Figures 4 and 5 have error bars and letters that seemingly denote statistical differences, but the caption is missing all the other requisite details.
9) The total flavonoid content of the samples is rather surprisingly high. Given that a hexane:ethanol mixture was used (according to Kaur et al.), it’s possible that some of the absorbance observed in the TFC assay (and by extension, many of the other aqueous-based assays in this study like the TPC assay) may have been contributed by precipitation of lipids in the system. This “precipitate” sometimes takes the form of a slimy, translucent material, which isn’t immediately obvious unless closely inspected but sufficient to throw off the accuracy of absorbance readings. It would also explain the larger-than-usual error bars.
Author Response
The study investigates the possibility of using thistle biomass to grow thistle mushrooms (Pleurotus eryngii), with the hypothesis essentially being that thistle makes thistle mushrooms grow better, possibly due to thistle’s chemical content (e.g. inulin). This involves studying the
growth and biochemical properties of the resultant mushrooms. The project conceptually makes sense, and the general straight-forwardness of the project’s intent and experimental design make for a well-contained and theoretically easily-understood.
 There are of course, certain things to address:
1. Inulin is a common compound, and certainly not unique to thistle; nor was it specified in the publication that thistle root has, for example, exceptionally high levels of inulin, that would justify selecting only inulin as a supplement. Given inulin’s widely-reported function
as a useful substrate for microbes/as a prebiotic, it isn’t surprising that it would have some impact on fungal growth as well.
In other words; what other compounds are present in thistle (either unique to thistle, or in exceptionally high levels) that could potentially be studied? Biotin and thiamine were mentioned in the introduction, but were not tested.
In our knowledge it is well established that thistle roots are rich in inulin up to 40% of dry matter as reported in Melilli et at 2019 and Pari et al 2021. The use of thistle roots biomass with its composition
(vitamins, compounds, and so on), allows to replace the use of molasses as carbon source. Molasses is principally a carbon source (saccarose), and normally it is obtained from abroad so that it should
be replaced to pursue circular economy.
We agree with the referee so we changed the graph with a table to clarify the significant results obtained from the interaction between P. eryngii isolates and substrates.
This revision has been added
2)      Not clear why the authors say that “Pleurotus eryngii is one of the most important cultivated mushroom species”. What is especially important about this species?
Our work is related to the Pleurotus cultivation in Italy. Moreover, P. eryngii cultivation is well developed in areas in which also thistle is cultivated and so we have large availability of its waste
biomass that allow us to transform waste into protein to pursue circular economy.
3)      For section 2.5, how long were the samples subjected to 105◦C? And how did the authors ensure that the sample was completely dried? E.g. were samples reweighed at least twice for consistency?
This revision has been added
4)      For the statistical analysis in Figure 2, presumably the authors are comparing between the five strains within each growth media (i.e. all three growth media are independently evaluated). Might also be worth analyzing for significant differences in growth for the same
strain but different growth media.
This revision has been added
5)      I personally have reservations about the use of only a triplicate when evaluating samples like this. Even though many have done it, in my personal experience having a larger number of replicates (e.g. five) gives more reliable information. This is especially so for when error bars
are quite large (e.g. some of the columns in Figures 4 and 5).
We amended the mistake reporting that the means derived from 3 experiments and three technical replicates. Secondary metabolites as well as bioactive compounds have high levels of variability among biological replicates. The aim of the work is to evaluate if new cultivation technique improves the content of bioactive compounds.
6)      Even within the same column, the number of significant figures/decimal points are sometimes inconsistent. For example, “7 ± 0.62”.
The revision has been added. The change was done according to the measurement of parameters
7)      The formatting of CaCO3 in the Table 3 legend is inconsistent.
This revision has been added
8)      Figure 3 is missing indicators of replication, error bars, and statistical analyses. Figures 4 and
5 have error bars and letters that seemingly denote statistical differences, but the caption is missing all the other requisite details.
In Figures 4 and 5 all details have been added.
9)      The total flavonoid content of the samples is rather surprisingly high. Given that a hexane:ethanol mixture was used (according to Kaur et al.), it’s possible that some of the absorbance observed in the TFC assay (and by extension, many of the other aqueous-based
assays in this study like the TPC assay) may have been contributed by precipitation of lipids in the system. This “precipitate” sometimes takes the form of a slimy, translucent material, which isn’t immediately obvious unless closely inspected but sufficient to throw off the
accuracy of absorbance readings. It would also explain the larger-than-usual error bars.
Extraction of bioactive compounds has been carried out following the procedure as described by Kaur et al using ethanol (80%). We agree with all considerations, but all samples have been centrifuged at the end of TPC and TFC assay before to determine their absorbance.
In addition, all the revisions suggested in the manuscript have also been accepted.
Reviewer 3 Report
Reviewer
Potential use of thistle biomass on Pleurotus eryngii production: 3 heteroglycans content and nutritional properties
Valerio Battaglia*1 , Roberto Sorrentino*1 , Giulia Verrilli1,2, Luisa del Piano1 , Maria Cristina Sorrentino1 4 , Milena Petriccione3 , Mariarosaria Sicignano1 , Anna Magri3,4, Michele Cermola1 , Domenico Cerrato1 and Ernesto Lahoz1
Comment:
(1) The research is very interesting and valuable for cultivation of P. eryngii.
(2) I really appreciate that the authors have repeated the experiments three times (trials) before publishing. This is be commended! Too often research is published on only one trial.
(3) The English is good BUT the manuscript would GREATLY benefit to have a native English speaker (first language and schooled in English grammar etc) read it for smoothness and readability of the language. For example [original] L 53 “In Italy the most used carbon source to grow P. eryngii is molasses imported from abroad and containing saccarose. The affinity of P. eryngii to inulin allow us to replace molasses with thistle roots” would read more smoothly as [CHANGE] “In Italy the most used carbon source to grow P. eryngii is molasses imported from abroad BECAUSE IT IS RICH IN saccarose. The affinity of P. eryngii FOR inulin WOULD allow us to replace molasses with thistle roots”. NOTE the reader may understand the intent of the authors but this is not a professional presentation by the authors. Some sentences are wordy. This certainly is the challenge when manuscripts are published in language other than that of the authors.
(4) As a researcher, I would want to duplicate the experiment. There are significant details missing that prevent me from doing so. Please note my comments throughout.
(5) This and other items necessitate a major rewrite.
L 1 – 4 The affiliation are both in Italian (1, 3) and English (2, 4). It should be consistent unless this is the actual wordings at the respective institutions.
L 25 “Choosing …. product.” rewrite – not clear
L 34 – the scientific name of the “thistle” should be in Keywords
L 42 “thistle” - there are over 24000 species of thistles. I presume (it is not stated) that this species is used for the study. The word “cardunculus” is only used twice in text of manuscript. The word “thistle” is used 60 times in the manuscript. Is this the species for the entire discussion and every use of the word? This is a clarity issue. The authors need to educate the readers on this thistle species. There is info in Discussion but should be included here. Thistle conjures up, at least in the mind of this reviewer, weeds that are invasive and compete for pastureland for livestock.
L44 - Comment: Since the premise of the paper is to use a thistle (not identified) byproducts (wastes?) to grow Pleurotus, then some info on the commercial production of this variety and availability of its byproducts would be of great value to the manuscript. Are the byproducts used elsewhere? Are there readily available byproducts?
L 48 What are thistle “scraps”
L 56 “mushrooms” which mushroom or all?
L 59 “technique” no technique has been mentioned.
LL 56 – 89 - The context is relevant. HOWEVER, it does not read smoothly. It appears to be “cut and paste” from elsewhere. Please rewrite.
L 91 – WHICH thistle; see note L 42 earlier.
L 101 - Why the sequencing?
L 116 – spell out number at beginning of sentence.
L 120 – how many days?
L 124 – what variety of wheat? This makes a difference. Cannot duplicate experiment.
L 124 – wet or dry thistle stems and roots? Later we read about hydrating dried material.
L 126 - How much of stems and roots produced the C/N ratio? I can’t repeat the experiment.
L 141 – Why Pe1Ce isolate? Perhaps say “ … as determined by results from sec 2.3”.
L 152 - Nothing has been said about stems that were dehydrated?
L 152 ff – Why the 600 g of plant material?
L 156 - is the WSM the control (industry standard)?
L 157 – I realize that industry will soak and drain substrate. However, are valuable nutrients being discarded? In the mycelial growth portion of paper this was not done. This does not permit projection from mycelial growth on agar to substrate.
L 166 – how many days? Or perhaps this will be data collected for each substrate. Please tell the reader.
L 167 - “opened” - top, cuts or what? details please.
L 172 - How many bags for each substrate combination? Stats (sect 2.8) says three independent experiments in triplicate. Section 2.6 says two independent in triplicate. So … does this mean 3 bags per substrate which is 12 total bags? What was the statistical design for the actual growth of mushrooms? At what stage were fruiting bodies harvested? Trimmed? Etc. In line 186 there are 6 fungal blocks? There are details missing!!!
L 174 - Even though the authors have referenced a protocol, some detail is valuable. The referenced paper is different species and grows much differently than P. eryngii.
L 177 - ‘have been realized’ ???
L 183-186 --- this does not belong is this section.
L 193 - Is “TRS” same as “TSR”? why only two of the four substrates?
L 219 - How many samples? From each bag?? Three independent experiments?? We’re not told.
L 222 - See earlier notes relative to actual number of experimental units per trial. Stat design of Table 2 is 5 x 3 factorial (4 experimental units per combination) with repeat experiment. ALSO, no where other than here is there mention of the test for homogeneity of variance. The authors need to proof that the variances between the three trials (repeats of experiment) were indeed NS.
L 237 – Not clearly demonstrated in the table 2.
L 248 - I would add with “four replicates per experiment”. Also, nothing has been said about whether or not HOV test was significant or not. Rather than repeat this comment about HOV – it is true of all.
L 252 - was the sequencing just to confirm the species? Could not more be said?
L 259 - presumably Pe1Ce; Also, nothing has been said about whether or not HOV test was significant or not.
L 300 - Presumably reference 15.
L 307 - TRS or TSR
L 361, 381 – “in the last years” - not English phrase; try “in the last few years” or “recently” or “in the last decade etc”
L 363 – “aggiungere Pari et al 364 2021 e Mleilli et 2019” ?????
L 365 - “avoind” likely ‘avoid’; better wording = “replacing”
L 378 – “According to Carrieri et al. [13] we evaluated the” ???
L 400 - “mushrooms is amply variable, it depends on the” - unclear; also, the “,” at minimum should be a “.”.
Author Response
Dear Referee,
thanks a lot for all your valuable comments and suggestions.
We accepted all your indications except the following
L 157 – I realize that industry will soak and drain substrate. However, are valuable nutrients being
discarded? In the mycelial growth portion of paper this was not done. This does not permit
projection from mycelial growth on agar to substrate.
The assay carried out by the authors with cardunculus (steam or roots paragraph 2.3) had the aim to
measure the affinity of P. eryngii mycelial growth in order to obtain carpophores in less time than
other substrates used in mushroom industry. Indeed to obtain heteroglycans from P. eryngii
mycelium (paragraph 2.4), the method proposed in the present work is to grow mycelium in a liquid
phase (industrial production) without the addition of biomass, but using inulin from cardunculus as
carbon source.
Reviewer 4 Report
Battaglia and colleagues demonstrate Potential use of thistle biomass on Pleurotus eryngii production: heteroglycans content and nutritional properties, but several issues should be addressed.
1. Line 53 and 54: P. eryngii is not italic.
2. Line 122: P. eryngii is not italic
3. Line 195: what is the meaning of (GAE 1g DW)?
4. Line 197: Since author described catechin equivalent per 100 gm. So100 g-1 DW is not necessary.
5. Line 200: same as line 197.
6. Figures 3 and 4 need sharper graphs with the front size should large. So, it will be more convenient for the reader.
7. In discussion: all phenolic content expression should be similar. Some times author expressed as mgGAE g-1 DW and mgGAE/g DW. So, it is my suggestion author should use either mgGAE g-1 DW or mgGAE/g DW.
8. Conclusion part is missing. The author should include the conclusion section more concisely.
Author Response
Dear Referee,
thanks a lot for all your valuable comments and suggestions.
We accepted all your indications that are included in the new version of the manuscript
Round 2
Reviewer 1 Report
The authors did not comply with or respond to the necessary changes to the manuscript
Author Response

(The authors gave the same response as above.)
